# Know-do gap for sick child care and drivers of knowledge and practice among health extension workers in four regions of Ethiopia: a cross-sectional study

Dawit Wolde Daka [1], Muluemebet Abera Wordofa,[2] Mirkuzie Woldie[1]

[1]Department of Health Policy and Management, Jimma University, Jimma, Ethiopia
[2]Department of Population & Family Health, Jimma University, Jimma, Ethiopia

**Correspondence to**
Dawit Wolde Daka;
dawit.daka86520@gmail.com

## ABSTRACT

**Objective** Though efforts were made to expand community-based sick child healthcare in Ethiopia, the quality of care provided remained low. Improving quality of care requires understanding providers' knowledge of appropriate care and their actual execution of tasks. This study examined gap between what health extension workers (HEWs) knew and did during the management of sick children in Ethiopia.

**Design** Facility-based cross-sectional study was conducted.

**Setting** The study was carried out in 52 districts across 4 regions in Ethiopia.

**Participants** We interviewed 274 HEWs and performed observations of consultations done by 150 HEWs supplemented with facility assessment from December 2018 to February 2019.

**Outcome** We compared providers' knowledge and performance in the management of childhood pneumonia and diarrhoea. Know-do gap implies the difference in proportion between knowledge and actual practice of HEWs. Logistic regression was used to identify predictors of knowledge and actual practice.

**Results** Providers' correct knowledge ranged from 27.8% to 76.0% for signs and symptoms of pneumonia, and 32.0% to 84% for dehydration signs. Their actual practices ranged from 15.1% to 47.3% for pneumonia and 27.0% to 42.6% for dehydration. The correct knowledge and actual practices for pneumonia and dehydration management were 88.3% vs 15.6% and 93.9% vs 51.3%, respectively. There was significant know-do gap in assessments (16.7%, p=0.002) and management of childhood conditions (68.5%, p<0.0001). Mentorships were associated with providers' knowledge of clinical management, while medicines availability was associated with their actual management practice.

**Conclusions** While knowledge and actual practice for assessment and management of pneumonia and dehydration ranged from very low to high, what is more concerning is the huge know-do gap among HEWs. Our findings suggest that knowledge-based training is necessary but not sufficient for ensuring correct assessment and management of sick children by HEWs. Continuous support through mentorships and the supply of commodities are critically needed.

## STRENGTHS AND LIMITATIONS OF THIS STUDY

⇒ The study used a validated tool that was tested in various low-income and middle-income settings for the evaluation of the quality of sick child care.
⇒ Observations of sick child consultations were performed by clinicians who had received refresher training on integrated community case management.
⇒ Though sample size was small for the know-do gap analysis, the study has covered the four largest regions of the country.
⇒ The observation method used in this study may have led to the Hawthorne effect possibly masking the actual practice due to a change in the professional behaviour of the health extension workers observed.

## INTRODUCTION

Pneumonia and diarrhoea remain the two major childhood illnesses that contributed to the larger portion of under-5 mortality worldwide.[1 2] Globally, over 1.23 million children died of pneumonia and diarrhoea before reaching their fifth birthday in 2018. Seventy per cent of all pneumonia and diarrhoea deaths in children under the age of 5 years have occurred in 15 countries of the world alone, and Ethiopia is one among them. In Ethiopia, 44, 692 under-5 children's deaths were attributed to pneumonia and diarrhoea in 2017 which accounted for 13.6 deaths per 1000 live births. Out of sick children with suspected pneumonia, only 31% were taken to an appropriate care provider and 7% received the recommended antibiotics. Similarly, of those under-5 children with diarrhoea, 30% have received oral rehydration solution (ORS) and 33% were provided with zinc supplements.[3 4]

The integrated community case management (iCCM) approach to common childhood illnesses is a key strategy for improving the treatment of potentially life-threatening

illnesses in children.[5–7] The iCCM guideline recommends that all children 2–59 months of age are first assessed for general danger signs, then for cough or difficult breathing, diarrhoea, fever, ear problem, acute malnutrition and anaemia. Then children are classified based on assessment results and treated according to the standard of care.[8] Adherence to guidelines by front-line providers is critical to improving the quality of care for sick children and improving treatment outcomes.[9] However, studies have shown that providers' adherence to the clinical standards of sick children assessment and management is low in most low-income and middle-income settings[10–12] and a significant know-do gap in clinical practices was common among the healthcare providers.[13 14]

The know-do gap, failure to apply existing knowledge to improve health outcome, has a huge impact on the effort made towards reducing under-5 mortalities attributed to poor quality or substandard care. It results in inconsistencies in care and inefficiencies in workflows. Though much is known about the evidence-based interventions related to maternal and child healthcare globally, it is not practised as intended and the practices vary across the different settings.[15] Most of the know-do gap is a result of inadequately skilled providers, lack of key equipment and medicines, and financial resources.[14 16 17] Other factors impacting on the likelihood of providers practising what they know include provider background characteristics (sex, age, working experience, level of education), extent of community support and presence support in the form of regular trainings, mentoring and supportive supervision.[11 13 18–20] On the other hand, if community health workers are supported adequately, they can assess and treat sick children according to the protocol in the majority of the cases.[21–23]

Narrowing the know-do gap is critical to attaining child survival goals and the targets set under the Global Action Plan for the Prevention and Control of Pneumonia and Diarrhoea in 2025. One of the core interventions outlined in the action plan was the appropriate management of sick children who are at risk of serious illnesses or death due to these two diseases (pneumonia and diarrhoea). It is aimed to increase treatment coverage for children with suspected pneumonia including care by an appropriate healthcare provider and antibiotics, and treatment coverage for children with diarrhoea including treatments with ORS and zinc supplements to 90%.[4]

In Ethiopia, the iCCM services implementation was started in 2010 as part of the health extension programme.[21 24] Currently, the programme is implemented in more than 18 000 health posts nationally.[25] Despite huge investments and high coverage of clinical training,[21 26] studies revealed that the quality of care is low in the health posts of Ethiopia in terms of sick children assessment, classification, treatment and referral practices.[27–29] On the other hand, little is known about the know-do gap in sick child assessment and management, and the drivers among health extension workers (HEWs) in Ethiopia. Therefore, this paper aimed to examine the level of agreement between the knowledge the HEWs had about assessment and management and their actual practice of childhood pneumonia and diarrhoea.

## METHODS

### Study setting and period

The study was conducted in the health posts of 52 districts across the four most populous regions of Ethiopia: Oromia, Amhara, Southern Nations Nationalities and People's and Tigray from December 2018 to February 2019. The four regions and the districts were selected by the Ministry of Health, Regional Health Bureaus and the implementing partners in 2016 for the planned intervention. The intervention districts were selected for having poor primary health service utilisation indicators for under-5 children and the comparison districts were identified based on their similarity in population size and key maternal and child health indicators. This paper was part of the larger Dagu end-line survey, a survey that aimed to estimate the effectiveness of the optimising health extension programme (OHEP) in Ethiopia and the baseline survey was conducted from December 2016 to February 2017. The protocol for the OHEP evaluation was published elsewhere.[30]

### Study design and participants

A facility-based cross-sectional study was conducted among HEWs that fulfilled the inclusion criteria. Two-stage stratified sampling strategy was used to select the study participants from the districts. In the first stage, enumeration areas (EAs) were selected from the study districts using the list of EAs from the 2007 Ethiopian Housing and Population Census as the sampling frame. The cumulative population size to clusters across the study areas was calculated and 200 EAs were selected with a probability proportional to population size (PPS). Each EA formed a cluster and each cluster constituted a primary sampling unit. In the second stage, all health posts serving the catchment population of the selected EAs were included in the study. Similarly, all HEWs who were present during the survey period were included in the study. Moreover, the sick children who visited the selected health posts to seek care from HEWs at the time of data collection were included.

### Measurement

Structured interviews, observation and health post assessment questionnaires were used to collect data. The questionnaires were developed based on the WHO tool for Integrated Management of Childhood Illnesses sick child quality assessment in primary healthcare facilities[31] and the national iCCM guidelines.[32] The health provider's interview questions included items such as background characteristics, iCCM training, supportive supervision, clinical mentorships, and knowledge of assessment and management of sick children 2–59 months. The observation questionnaire captured data on the reason for visits,

assessment, classification, treatment, referral and counselling. The facility assessment tool had sections of health posts infrastructure, medical equipment, medicines and job aids. All the questionnaires were uploaded on tablets (CSProV. 7.1) for data collection.

Data collectors with a qualification of bachelor's degree in health science were recruited and involved in data collection. Likewise, supervisors with a health background and qualifications with a bachelor of science and above have participated. Data collectors and supervisors were trained for 10 days on the questionnaires, data collection techniques, how to use tablets and research ethics. Clinical observations of providers were conducted following interviews and facility assessments. The supervisors have monitored the daily data collection process at the field level and provided support. Moreover, a data manager at the central office monitored data on the server and provided feedback to the field team.

## Study variables

The outcome variable was the know-do gap of child care related to pneumonia and diarrhoea assessment and management. The providers' knowledge data were obtained from interviews, and the actual practice data were collected through observation by clinicians. Know-do gap was the difference between the knowledge of HEWs about the signs and symptoms and management of pneumonia and diarrhoea and their ability to correctly assess those signs and symptoms when children are presented with the illnesses. A provider was regarded as having a know-do gap in the following conditions: (A) if correctly named the signs and symptoms of the illnesses but failed to actually identify them; (B) if unable to correctly named the signs and symptoms of the illnesses but able to identify them and (C) if unable to correctly named the signs and symptoms of the illnesses as well as unable to correctly identify them. On the other hand, if the provider correctly named the signs and symptoms, and also correctly identified them, then he/she was regarded as with no know-do gap. Similarly, the above conditions were applied to the know-do gap of pneumonia and dehydration management by the providers.

A total of 11 knowledge items were used, each having 'yes=1' and 'no=0' responses. Providers were asked to respond to questions without prompts and the responses were recorded from the predefined list. Then the interviewer recorded the responses of the providers. The knowledge questions were: what are the main signs and symptoms of pneumonia in children 2–59 months (three responses)? When a child presents with signs and symptoms of pneumonia, what initial measures do you take (two responses)? What are the main signs and symptoms of diarrhoea in children 2–59 months (four responses)? When a child presents with signs and symptoms of diarrhoea, what initial measures do you take (two responses)?. The initial measures taken by providers for pneumonia have included prescribing antibiotics and counselling

on medications, and for diarrhoea, the actions were providing ORS and zinc, and counselling on medications.

We have constructed the knowledge index based on the Ethiopian university students' grading system with a few modifications as there was no validated standard to evaluate providers' knowledge. The Ethiopian university grading system is excellent (85%–100%), very good (75%–85%), good (60%–75%), satisfactory (50%–60%), unsatisfactory (45.50%) and very poor (below 45%). The providers who have scored 60% and above were categorised as having 'good knowledge/knowledgeable=1', and otherwise 'poor knowledge/not knowledgeable=0'.

Hence, the HEWs were labelled knowledgeable if they have correctly responded to ≥60% of the knowledge items (3 and more items out of 5 for pneumonia; 4 and more items out of 6 for diarrhoea; 7 and more items out of 11 for average knowledge). All scores below 60% were regarded as poor knowledge for the individual and average scores.

The ability to correctly assess and manage children (the actual practice of providers) for pneumonia and diarrhoea index was constructed. The providers who correctly assessed children that presented with respiratory and diarrhoea complaints according to the iCCM guidelines were categorised as 'correct practice for assessment=1' and otherwise 'incorrect practice for assessment=0'. Likewise, the providers who correctly treated children who have been classified as having pneumonia or diarrhoea at the outset according to the iCCM guidelines were categorised as 'correct practice for treatment=1' and otherwise 'incorrect practice for treatment=0'. The providers were observed while consulting more than one sick children and hence, correct assessment and treatment for both conditions were defined as being able to correctly assess and treat all children according to the iCCM guideline.

The tasks of pneumonia assessment according to the iCCM guidelines were 'ask for the presence of cough or difficult breathing, check chest-indrawing and listen for stridor'. The tasks for diarrhoea assessment were 'ask for the presence of diarrhoea, ask the presence of blood in the stool, check child restlessness or irritability, and check whether skin pinch goes back slowly or very slowly'. The task of pneumonia treatment was providing antibiotics to sick children classified as having pneumonia, and the task of diarrhoea treatment was providing ORS and zinc to those sick children who were classified as having diarrhoea. The providers that correctly assessed children for pneumonia and diarrhoea were categorised as 'correct practice for assessment=1' and otherwise 'incorrect practice for assessment=0'. Those providers who treated all children who were classified as having pneumonia, and diarrhoea with antibiotics, and ORS and Zinc were labelled as 'correct practice for treatment=1' and the providers who did not provide recommended treatments in either of the conditions were labelled as 'incorrect practice for treatment=0'.

The explanatory variables of knowledge and actual practices were provider characteristics (age, working

experience as HEWs, training level of HEWs (level-3 and level-4)), iCCM training status, presence of supportive supervision and mentorships, and presence of medicines and supplies. Level-3 HEWs are those who graduated with a certificate after a 1-year preservice training on the 17 packages of the health extension programme, and level-4 HEWs are those who have level-3 certificate and got one additional year training at a technical college.[25 33] The 17 packages of the health extension programme are categorised into four components or areas as (A) family health services (maternal and child health, family planning, immunisation, adolescent reproductive health, nutrition); (B) disease prevention and control (HIV/AIDS, tuberculosis, malaria, first aid); (C) hygiene and environmental sanitation (proper and safe excreta disposal, proper and safe solid and liquid waste management, water supply safety measures, food hygiene and safety measures, healthy home environment, arthropods and rodents control, personal hygiene) and (D) health education and communication.[25]

## Data processing and analysis

Primarily, the HEWs module data set and the observation module data set were linked using individual ID and then linked to the health facility assessment data set. The health facility assessment data included information on health post infrastructure, availability of medical equipment, medicines and job aids. Data values were cleaned for inconsistency and incompleteness. Coding and recoding of variables were done, and the indices were computed as appropriate. The know-do gap analysis was conducted for those HEWs who have been observed while consulting sick children, and analysis for knowledge about sick child management was conducted for all the HEWs included in the study. The knowledge analysis was conducted on 274 HEWs, and from these the know-do gap analysis was performed with 150 HEWs who have consulted sick children on the date of the survey. Due to the smaller sample size, we were unable to do regression for know-do gap and preferred to perform a separate model for knowledge and practice of the HEWs who did sick child consultations.

Descriptive statistics were performed to analyse and describe the characteristics of providers, other explanatory variables and the outcome variable. A 95% CI was constructed for the outcome variable and some of the explanatory variables. To assess whether there exist significant differences between the proportion of providers with the correct knowledge and correct practice, we have conducted two-sample z-tests for differences in proportions. Bivariate logistic regression was used to select candidate explanatory variables for multiple variable logistic regression at $p < 0.2$ and multiple variable logistic regression was used to analyse the association between explanatory variables with knowledge, and actual management practice of the providers. The variables considered in this analysis were provider age, years of experience, level of education (level-3 and level-4), iCCM training status, presence of supportive supervision and mentorships, and

presence of medicines and supplies. Those variables with a $p < 0.05$ were considered statistically significant. All analyses were performed by using SPSS V.20 (IBM).

## Patient and public involvement

Patient and/or the public were not involved in the design or conduct, or reporting or dissemination plans of this research.

## RESULTS

### Description of background characteristics

From the total of 274 HEWs interviewed, 150 examined sick children and were included in the know-do gap and actual practice analysis. While 274 HEWs were used in the knowledge analysis. The majority (78.8%) of providers had received iCCM training and out of which the larger proportion had received the training within the 12 months preceding the date of the survey (187 or 86.6%). Though the majority of the HEWs had received supervision within the 3 months preceding the survey date (71.9%), only 33.9% had been visited regularly. Furthermore, less than half (46%) of the HEWs participated in the performance review clinical mentoring meetings (PRCMM) that were held 6 months preceding the survey date. The details for the background characteristics of the HEWs included in the study are provided in table 1.

The HEWs have consulted a total of 685 sick children, with an average of 5 sick children consultations per HEW (SD 1.6). The minimum and maximum consultations were 1 and 6, respectively, and the majority, 127 (84.7%), of the HEWs have consulted three or more sick children. Out of the total sick children who visited the HEWs, 628 (91.7%) were older than 6 months. Three hundred sixteen (46%) sick children had respiratory complaints and 302 (44%) had diarrhoea related complaints.

### Medical equipment, job aids and medicines availability

A total of 121 (82.3%) providers had thermometers, 115 (78.2%) had infant scales, 116 (78.9%) had weighing slings, 79 (53.7%) had tape measures, 69 (46.9%) had blood pressure (BP) apparatus, 98 (66.7%) had stethoscopes, 16 (10.9%) had watch or clock and 102 (69.4%) had examination couch. Whereas all of the providers reported that they had mid-upper arm circumference tape, which is used to assess malnutrition. Regarding job aids and tools, 145 (98.6%) had iCCM register for 2–59 months old children, and 138 (93.9%) had chart booklets.

Overall, 115 HEWs (71.4%) out of 147 had access to amoxicillin on the date of assessment. ORS medicines were available to 137 (93.2%) providers and zinc to 125 (85%) providers. Both ORS and zinc medicines were available to 117 (79.6%) providers (online supplemental table S1).

### Providers knowledge

Out of 274 providers who responded to the interview (online supplemental table S2), less than half of them

**Table 1** Background characteristics of the health extension workers in four regions of Ethiopia, December 2018–February 2019

| Variables | n (%) |
| --- | --- |
| Age in years | |
| ≤30 | 187 (68.2) |
| >30 | 87 (31.8) |
| Level of training* | |
| Level-3 (certificate) | 154 (56.2) |
| Level-4 (diploma) | 120 (43.8) |
| Work experience as an HEW in years | |
| ≤5 | 106 (38.7) |
| 6–10 | 63 (23.0) |
| >10 | 105 (38.3) |
| iCCM training | |
| Yes | 216 (78.8) |
| No | 58 (21.2) |
| Received supportive supervision in the last 3 months | |
| Yes | 197 (71.9) |
| No | 77 (28.1) |
| No of visits (in the last 3 months) | |
| No visit | 77 (28.1) |
| 1–2 | 104 (38.0) |
| 3 and more | 93 (33.9) |
| Contents of supportive supervision† | |
| Discussing diagnosis and treatment of suspected pneumonia | 132 (67.0) |
| Discussing diagnosis and treatment of suspected diarrhoea | 139 (70.6) |
| Observing record keeping | 187 (94.9) |
| Checking the register for consistency and completeness | 183 (92.9) |
| Checking supplies including manuals, job aids and request forms | 146 (74.1) |
| Delivering supplies including manuals, job aids and request forms | 123 (62.4) |
| Observing client consultation with providers | 99 (50.3) |
| Providing written feedback | 131 (66.5) |
| Participated in PRCMM in the last 6 months | |
| Yes | 126 (46.0) |
| No | 148 (54.0) |
| Region | |
| Amhara | 130 (47.4) |
| Oromia | 71 (25.9) |
| SNNP | 37 (13.5) |
| Tigray | 36 (13.1) |

*Level of training: HEW level of training categories: level-3 HEWs are those who graduated with a certificate after a 1-year preservice training on the 17 packages of the Health Extension Programme. Level-4 HEWs are those who graduated with a diploma after 1 year of additional education at a technical college.
†Out of the total health extension workers that have received supervision in the last 3 months (N=197)
HEW, health extension worker; iCCM, integrated community case management; PRCMM, performance review clinical mentoring meeting; SNNP, Southern Nations, Nationalities and Peoples.

had good knowledge of pneumonia signs and symptoms (35.4%), diarrhoea signs and symptoms (20.8%), pneumonia management approaches (24.5%), and diarrhoea management approaches (24.1%). Overall, slightly over a quarter (27%) of the providers were knowledgeable about the signs and symptoms, and management approaches of both illnesses. Mean pneumonia and diarrhoea signs and symptoms, and the management knowledge score of the HEWs was 48.31% (95% CI 45.7% to 50.9%) with an SD of 21.5.

### Know-do gap in child healthcare

There was a significant difference between the knowledge of the HEWs and their actual practices in the assessment of pneumonia and diarrhoea ($p < 0.05$ for individual and overall know-do gap proportions as in table 2). Slightly greater than one-third (35.7%) of HEWs had good knowledge of pneumonia signs and symptoms and only one-fifth of them (19.8%) had actually assessed those signs and symptoms. Though three-fourths of providers know cough or fast breathing as the sign and symptoms of pneumonia, only 47.3% of them have actually asked about cough or difficulty breathing. Likewise, 34.9% of providers have described chest-indrawing as a symptom of pneumonia while only 28.6% have assessed chest-indrawing in children.

Six out of 10 (61.5%) of the providers have good knowledge of diarrhoea signs and symptoms, however only 4 out of 10 (40.2%) were able to actually assess those signs and symptoms in children who presented with diarrhoea complaints.

Similarly, a significant difference was observed between the knowledge and actual practice of providers regarding the treatment of childhood pneumonia and diarrhoea ($p < 0.05$). Most of the providers (96.2%) have reported that they knew treatment recommendations for pneumonia and diarrhoea, but only a little more than a quarter of them (27.7%) have actually practised the recommendations for children who were classified as having either pneumonia or diarrhoea. On the contrary, a large proportion of providers practised counselling about medications for children who were provided treatments for pneumonia and diarrhoea though only a few of them know it as a component of management (table 3).

### Drivers of knowledge and actual practice of providers
#### Drivers of providers' knowledge

The majority of the providers who have service experience greater than 5 years (74.3%) as HEWs and who received iCCM training (83.8%) had good knowledge of pneumonia and diarrhoea signs and symptoms and management approaches. Moreover, 6 out of 10 of the providers who have received supervision with components of discussing diagnosis and treatment of childhood illnesses and those who participated in the PRCMM had good knowledge.

In the bivariate analysis, providers' age, service experience, level of education, iCCM training, supportive

**Table 2** Providers know-do gap in the assessment of pneumonia and diarrhoea in four regions of Ethiopia, December 201–February 2019

| Assessment | Knowledge (n, %) | Practice (n, %) | Gap (%) | Z test p value |
|---|---|---|---|---|
| Pneumonia | | | | |
| Cough or fast breathing | 114 (76.0) | 71 (47.3) | 28.7 | p<0.0001 |
| Chest-indrawing* | 44 (34.9) | 36 (28.6) | 6.3 | 0.20 |
| Stridor* | 35 (27.8) | 19 (15.1) | 12.7 | 0.006 |
| Pneumonia index* | 45 (35.7) | 25 (19.8) | 15.9 | 0.005 |
| Diarrhoea | | | | |
| Presence of diarrhoea | 126 (84.0) | 62 (41.3) | 42.7 | p<0.0001 |
| Blood in the stool† | 39 (32.0) | 52 (42.6) | 10.6 | 0.17 |
| Child restlessness or irritability† | 39 (32.0) | 46 (37.7) | 5.7 | 0.26 |
| Pinch the skin of the abdomen† | 56 (45.9) | 33 (27.0) | 18.9 | 0.001 |
| Diarrhoea index† | 75 (61.5) | 49 (40.2) | 21.3 | 0.001 |
| Overall index‡ | 41 (38.0) | 23 (21.3) | 16.7 | 0.002 |

*Out of the health extension workers that have consulted children who were labeled to have a cough or difficulty breathing (n=126).
†Out of the health extension workers that have consulted children who were labeled to have diarrhoea (n=122).
‡Out of the health extension workers that have consulted children who were labeled to have both cough and diarrhoea (n=108).

supervision and participation on PRCMM were identified as candidate variables for multiple variable logistic regression at p<0.2. In the multiple variable logistic regression, only the participation in the PRCMM showed statistically significant associations with the knowledge of the providers. Providers' knowledge of pneumonia and diarrhoea management was nearly two times more likely among providers who were participated in the PRCMM that was held 6 months preceding the survey period (adjusted OR, AOR 2.00, 95% CI 1.09 to 3.67) (table 4).

### Drivers of providers' actual practice

The majority of HEWs aged less than 30 years (77.8%) and those who have access to medicines for childhood conditions (83.3%) had better childhood pneumonia and diarrhoea management practice. Moreover, greater than half

of the HEWs who received supervision with a component of discussing the management of childhood conditions, and less than half of the providers who participated in the PRCMM had good actual management practice.

In the bivariate analysis, providers' age, service experience, level of education, iCCM training, supportive supervision, participation on PRCMM, knowledge of childhood pneumonia and diarrhoea management, and availability of medicines were identified as candidate variables for multiple variable logistic regression at p<0.2. The multiple variable logistic regression analysis revealed that pneumonia and diarrhoea medicines availability was associated with the actual management practices of providers. The rest of the variables did not show associations. Correct management practice was four times more

**Table 3** Providers know-do gap in the management of pneumonia and diarrhoea in four regions of Ethiopia, December 2018–February 2019

| Treatment and counselling | Knowledge (n, %) | Practice (n, %) | Gap (%) | Z test p value |
|---|---|---|---|---|
| Pneumonia | | | | |
| Prescribe antibiotics* | 68 (88.3) | 12 (15.6) | 72.7 | p<0.0001 |
| Counsel on medications† | 2 (16.7) | 10 (83.3) | (66.6) | 0.001 |
| Diarrhoea | | | | |
| Prescribe ORS and zinc‡ | 108 (93.9) | 59 (51.3) | 42.6 | p<0.0001 |
| Counsel on medications§ | 18 (30.5) | 30 (50.8) | (20.3) | 0.03 |
| Treatment for pneumonia and diarrhoea¶ | 126 (96.2) | 36 (27.7) | 68.5 | p<0.0001 |

*Out of the health extension workers who have consulted children classified as having pneumonia (N=77).
†Out of the health extension worker who has provided treatment to pneumonia (N=12).
‡Out of the health extension workers who have consulted children classified as having diarrhoea (N=115).
§Out of the health extension workers who have provided treatment to diarrhoea (N=59).
¶Out of the health extension workers who have consulted children who were classified as having either pneumonia or diarrhoea (N=130).
ORS, oral rehydration solution.

**Table 4** Correlates of providers' knowledge of pneumonia and diarrhoea management in four regions of Ethiopia, December 2018–February 2019

| Variables | Knowledge | | AOR (95% CI) |
| --- | --- | --- | --- |
| | Good (n,%) | Poor (n,%) | |
| Observations | 74 | 200 | |
| Age in years | | | |
| ≤30 | 43 (58.1) | 144 (72.0) | Ref |
| >30 | 31 (41.9) | 56 (28.0) | 1.41 (0.75 to 2.64) |
| Service experience as HEWs in years | | | |
| ≤5 | 19 (25.7) | 87 (43.5) | Ref |
| >5 | 55 (74.3) | 113 (56.5) | 1.78 (0.81 to 3.92) |
| Level of HEWs | | | |
| Level-3 (certificate) | 34 (45.9) | 120 (60.0) | Ref |
| Level-4 (diploma) | 40 (54.1) | 80 (40.0) | 1.33 (0.74 to 2.39) |
| iCCM training | | | |
| No | 12 (16.2) | 46 (23.0) | Ref |
| Yes | 62 (83.8) | 154 (77.0) | 0.71 (0.29 to 1.71) |
| Supervision with discussion on diagnosis and treatment of pneumonia and diarrhoea | | | |
| No | 29 (39.2) | 103 (51.5) | Ref |
| Yes | 45 (60.8) | 97 (48.5) | 1.12 (0.60 to 2.11) |
| Supervision with a component of observing client consultations | | | |
| No | 38 (51.3) | 131 (65.5) | Ref |
| Yes | 36 (48.7) | 69 (34.5) | 1.36 (0.73 to 2.53) |
| Being participated in PRCMM | | | |
| No | 29 (39.2) | 119 (59.5) | Ref |
| Yes | 45 (60.8) | 81 (40.5) | 2.00 (1.09 to 3.67)* |

*p<0.05 is considered statistically significant in the multivariate analysis.
AOR, adjusted OR; HEW, health extension worker; iCCM, integrated community case management; PRCMM, performance review clinical mentoring meeting; Ref, reference category.

likely in providers who have access to medicines for the treatment of pneumonia and diarrhoea compared with those who did not have access (AOR 4.04, 95% CI 1.48 to 11.06) (table 5).

## DISCUSSION

The study revealed that a low proportion of the providers had knowledge of the signs and symptoms and management of pneumonia and diarrhoea. Similarly, a low proportion of the providers executed the recommended actions for the assessment and management of childhood pneumonia and dehydration. There was a significant difference between the knowledge and actual practice of the providers. This finding is consistent with a result among other health professionals in Ethiopia[14] and frontline health workers in India.[13 16]

Slightly greater than one-third (35.7%) of the HEWs had good knowledge of pneumonia signs and symptoms and only one-fifth of them (19.8%) had actually assessed consulting children for those signs and symptoms. The widest gap was observed between knowledge and actual practice of providers related to cough or difficulty breathing (76% vs 47.3%) and stridor (27.8% vs 15.1%). Besides, the incongruity between the knowledge and actual practices of the providers; the level of concordance with the clinical guidelines have remained low among the frontline providers. This has a huge impact on the clinical care outcomes related to childhood pneumonia. Poor adherence to clinical guidelines by providers in the assessment of sick child was also reported in earlier studies from low-income and middle-income countries[34 35] and in Ethiopia.[27–29]

Though 6 out of 10 (61.5%) of the providers had good knowledge of signs and symptoms of diarrhoea, only 4 out of 10 (40.2%) actually assessed those signs and symptoms in children who presented with diarrhoea complaints. A wide and significant gap was observed between the knowledge and actual practice of the providers related to overall diarrhoea assessment (84% vs 41.3%) and pinch skin of the abdomen (45.9% vs 27%). On the contrary,

**Table 5** Correlates of the providers' actual management practice of pneumonia and diarrhoea in four regions of Ethiopia, December 2018–February 2019

| Variables | Practice | | AOR (95% CI) |
|---|---|---|---|
| | Good (n,%) | Poor (n,%) | |
| Observations | 36 | 94 | |
| Age in years | | | |
| ≤30 | 28 (77.8) | 70 (74.5) | Ref |
| >30 | 8 (22.2) | 24 (25.5) | 0.61 (0.22 to 1.75) |
| Service experience as HEWs in years | | | |
| ≤5 | 14 (38.9) | 39 (41.5) | Ref |
| >5 | 22 (61.1) | 55 (58.5) | 1.54 (0.59 to 4.02) |
| Level of HEWs | | | |
| Level-3 (certificate) | 20 (55.6) | 59 (62.8) | Ref |
| Level-4 (diploma) | 16 (44.4) | 35 (37.2) | 1.52 (0.62 to 3.68) |
| iCCM training | | | |
| No | 30 (83.3) | 75 (79.8) | Ref |
| Yes | 6 (16.7) | 19 (20.2) | 1.45 (0.39 to 5.30) |
| Supervision with discussion on diagnosis and treatment of pneumonia and diarrhoea | | | |
| No | 15 (41.7) | 42 (44.7) | Ref |
| Yes | 21 (58.6) | 52 (55.3) | 1.13 (0.44 to 2.87) |
| Supervision with a component of observing client consultations | | | |
| No | 23 (63.9) | 58 (61.7) | Ref |
| Yes | 13 (36.1) | 36 (38.3) | 0.91 (0.37 to 2.19) |
| Being participated in PRCMM | | | |
| No | 19 (52.8) | 46 (48.9) | Ref |
| Yes | 17 (47.2) | 48 (51.1) | 0.60 (0.24 to 1.50) |
| Knowledge index | | | |
| Poor knowledge | 25 (69.4) | 67 (71.3) | Ref |
| Good knowledge | 11 (30.6) | 27 (28.7) | 1.04 (0.41 to 2.64) |
| Amoxicillin, ORS and zinc availability† | | | |
| No | 6 (16.7) | 38 (40.4) | Ref |
| Yes | 30 (83.3) | 53 (59.6) | 4.04 (1.48 to 11.06)* |

*p<0.05 is considered statistically significant in the multivariate analysis.
†Out of providers who have assessed for medications availability in the health posts (n=127). Providers who had access to all the three medicines were regarded as 'yes' and otherwise 'no'.
AOR, adjusted OR; HEW, health extension worker; iCCM, integrated community case management; PRCMM, performance review clinical mentoring meeting; Ref, reference category.

a large proportion of providers have practised assessing the presence of blood in the stool (42.6% vs 32%), and child restlessness or irritability (37.7% vs 32%) without reporting on knowledge of the symptoms at the outset. Though the clinical guidelines recommend assessment of all the signs and symptoms of dehydration, the providers did not actually practice them.[32] This could lead to incorrect diagnosis of sick children with possible dehydration and results in missed opportunity to provide life-saving treatment for those in need of it.

The know-do gap analysis also revealed that there was a significant difference between the knowledge and actual practice of providers. Most of the providers (96.2%) have reported that they know recommended treatment options for pneumonia and diarrhoea, but only a quarter of them (27.7%) have actually provided treatment for children who were classified as having either pneumonia or diarrhoea. Earlier studies from Ethiopia and Uganda also revealed that community health workers had poor knowledge and practice of childhood illnesses management.[11 18 36] Contrarily, a large proportion of providers practised counselling about medications for children who were provided treatment for pneumonia and diarrhoea though only a few of them knew it as a component of sick child care.

Many factors counter play the lack of knowledge and good practice related to the assessment and management of sick children. The quality of clinical care provided to sick children is a function of the providers' knowledge, the effort exerted by the provider to apply the knowledge, and the readiness of health facilities to provide quality care in terms of medicines and supplies. In the current study, the providers' knowledge and actual practice of assessment and management of childhood pneumonia and diarrhoea were associated with participation in PRCMM and the availability of medicines for pneumonia and dehydration treatment at the health posts. A previous study also revealed that clinical coaching is positively associated with good clinical practice among community health workers.[37] The odds of good knowledge of pneumonia and diarrhoea assessment and management was two times more likely in the providers who participated in the PRCMM. While, the odds of better correct practice were four times more likely in providers who had medicines for the treatment of childhood pneumonia and diarrhoea (amoxicillin, ORS and zinc). Other variables including the providers' characteristics (age, work or practice experience, level of training), receiving clinical training and supportive supervision did not affect knowledge and actual practice of the HEWs. Though most of the providers were trained in iCCM, they scored low in the knowledge and practice for assessment and treatment of childhood pneumonia implying that in-serving trainings are necessary but not sufficient drivers of correct practice.[38] In another study, variables including the providers' level of education, work or practice experience, the presence of supervision and guidelines were associated with the community health workers' knowledge of childhood illness management.[18 36]

The findings of this study have policy and practice implications. Clinical training interventions that lacked adequate practical skill component do not improve the competency of providers. The lack of refresher training is also a possible cause for poor performance due to the decay in knowledge. High-quality in-service training combined with mentoring, supervisions, medicines and material support are effective to improve the quality of care.[38–40] Training had high coverage in the study area, but significant number of providers are short of iCCM medicines, supplies and technical support. Supervisions and mentoring focused on the management of childhood pneumonia and diarrhoea were limited. The supervisors discussed the management of pneumonia and diarrhoea in half of their visits (51.8%) and in greater than one-third (38.3%) they observed client consultations by the provider. Less than half (46%) of providers were clinically mentored according to the standard guideline, and two-thirds (65.7%) of them had access to medicines for pneumonia and diarrhoea. In the study area, in-service training and supervision were not associated with knowledge, and actual practices of providers signalling a look at other approaches that improves their quality.

The study was conducted in the community health posts within the primary healthcare level, the lowest care level in the health system of Ethiopia. At this level, the providers are supposed to diagnose and treat moderate cases, and refer severe cases to the health centres. This is the first study that examined the know-do gap in childhood pneumonia and diarrhoea care among HEWs in Ethiopia. The know-do gap analysis was based on observations of actual management practices of the providers by experienced and trained health professionals, and interviews on the knowledge assessment components. Though our findings are limited by a small sample size, the study has covered the four most populous regions of the country and is representative of the different settings in rural Ethiopia. The small sample size might have caused underestimated standard errors. On average, the providers were observed while consulting five sick children, and this might ensure the heterogeneity of cases for both conditions. However, the clinical practices and actions of providers related to severe illnesses was not examined in the present research. The other limitation might be the Hawthorne effect, that is, a change in professional behaviour of the HEWs when being observed.[41] This might partly mask the actual practice of providers. But, the findings related to actual management practice in both conditions shown that the providers have performed less and in normal management situation this might be even worse.

## CONCLUSIONS

The study revealed a huge know-do gap in the assessment and management of childhood pneumonia and diarrhoea among the HEWs in Ethiopia. The finding affirm that knowledge-based training is necessary but not sufficient to influence the clinical performance (actual practice) of providers. The know-do gap of pneumonia and diarrhoea management was determined by the presence of clinical mentoring and performance reviews and the availability of medicines. Therefore, an effort should be made to close the gap between theoretical knowledge and practice. The HEWs should be supported with mentoring, performance reviews, and the supply of medicines and commodities.

**Acknowledgements** The authors acknowledge the administrative hierarchies and the research participants in the study area for their cooperation throughout the research process, and the time they devoted to the study in providing required information.

**Contributors** All authors made a significant contribution to the work reported, whether, that is, in the conception, study design, execution, acquisition of data, analysis and interpretation, or in all these areas. DWD wrote the first draft of the manuscript, and MAW and MW revised the manuscript. All authors gave final approval of the version to be published, have agreed on the journal to which the article has been submitted and agreed to be accountable for all aspects of the work. All authors read and approved the final manuscript. DWD approved the final version to be published and accepts full responsibility for the finished work and/or the conduct of the investigation, had access to the data and controlled the decision to publish.

**Funding** This project was funded by Bill & Melinda Gates Foundation(INV-009691).

**Competing interests** None declared.

**Patient and public involvement** Patients and/or the public were not involved in the design, or conduct, or reporting, or dissemination plans of this research.

**Patient consent for publication** Consent obtained from parent(s)/guardian(s).

**Ethics approval** Research ethical approval was obtained from the Ethiopian Public Health Institute (protocol number SERO-012-8-2016; 001 August 2016), the London School of Hygiene and Tropical Medicine (protocol number 11235, June 2016), and Jimma University (protocol number IHRPGD/472/2018, August 2018). A support letter was secured from the regional health bureaus and zonal health offices. Informed written consent was taken from providers and for observation, consent was also taken from caregivers. For caregivers who were unable to read and write, witness was presented and signatures appended on the consent form. The confidentiality of providers and patient information was maintained properly through the use of codes and computer-based data were secured with passwords. Data will be used for the intended research purpose and will not be shared with a third party.

**Provenance and peer review** Not commissioned; externally peer reviewed.

**Data availability statement** Data are available on reasonable request. The datasets used and/or analysed for the current study are not openly available because data are part of the ongoing research project and available from the corresponding author upon reasonable request (email: dawit.daka86520@gmail.com; or dawit.daka@ju.edu.et).

**ORCID iD**
Dawit Wolde Daka http://orcid.org/0000-0001-5465-6345

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
