## [Reviewer comments · BMJ Open]

ARTICLE DETAILS

TITLE (PROVISIONAL)	Know-do gap for sick child care and drivers of knowledge and practice among health extension workers in four regions of Ethiopia: a cross-sectional study
AUTHORS	Daka, Dawit; Wordofa, Muluemebet; Woldie, Mirkuzie

VERSION 1 – REVIEW

REVIEWER	Dias, Carlos National Institute of Health Ricardo Jorge, Department of Epidemiology
REVIEW RETURNED	13-Dec-2022

GENERAL COMMENTS	Thank you for submitting the paper "Know-do gap for sick child care among health extension workers in Ethiopia" to be published in BMJ Open. This paper deals with a very relevant topic and I praise you for that. I enjoyed reading the paper and I hope that you may find my comments and suggestion useful for improving the document. The first suggestion is that you could complete the abstract by including information on the materials and methods used. These are described in the main text but not in the abstract, except a brief mention in lines 54 and 55. Also in the abstract but indeed throughout the document acronyms are used without first being used in full. For instance HEWs (lines 29, 84). In the main text, the materials and methods section can benefit from more detail. Sampling criteria (lines 102) for choosing the regions could be explicated. In Line 133 you should clarify if the difference you mention is the difference in proportions or in scores? Lines 136 through 142 describe the criteria for classifying HEWs as having a know-do gap or not. I am not sure to have completely understood the relation between this and the data from direct observation of caregivers. Could you please make this clearer in the document? Line 150: can you please clarify the "measures" that are mentioned? Line 151: Can you please clarify the reason for using the Ethiopian University Index and cut-offs? Lines 156 to 159: can you clarify the criteria for "correct practice" and "correct management" and "correct treatment". Line 173: should describe the "health facility assessment data set". Lines 175 to 181: should detail the logistic regression model namely variables used for adjustment and in the results section you present adjusted Odds Ratios. Lines 186 and 187: Did all participants provide a written consent? How did you manage those participants that could not read nor write?
--

	Lines 195 to 197: not all participants provided data on practice and knowledge. Can you specify these results? Line 199: please clarify "level-3 training", despite this is described in Table 1 but as a footnote. Line 208 - Table 1: The column heading "frequency" should be replaced by "number" or "n", as both columns represent frequencies (absolute and relative). I suggest that the discussion of results could include and discuss the role of other explanatory factors such as years of practice, severity of cases, and organization of care, which may be confounding the obtained associations. I wish you all the best for your work.
--	---

REVIEWER	Susihono, Wahyu Sultan Ageng Tirtayasa University
REVIEW RETURNED	19-Dec-2022

GENERAL COMMENTS	The reviewer has no comments.
-------------------------------

REVIEWER	Babu, Jobi Marian College Kuttikkanam, School of Social Work
REVIEW RETURNED	18-Jan-2023

GENERAL COMMENTS	 - Use APA 7th version for the references - Highlight the interventions proposed - Mention the limitations of the study
--

VERSION 1 – AUTHOR RESPONSE

REVIEWER 1:

Reviewer: Dear Prof Daka, thank you for submitting the paper "Know-do gap for sick child care among health extension workers in Ethiopia" to be published in BMJ Open. This paper deals with a very relevant topic and I praise you for that. I enjoyed reading the paper and I hope that you may find my comments and suggestion useful for improving the document. The first suggestion is that you could complete the abstract by including information on the materials and methods used. These are described in the main text but not in the abstract, except for a brief mention in lines 54 and 55. Also in the abstract but indeed throughout the document acronyms are used without first being used in full. For instance, HEWs (lines 29, 84).

Response: Thank you a lot for the insightful feedback and appreciation. We have added some information to make clear the methods and materials in the abstract section (Lines # 30 to 34). Due to the word limit, we are unable to describe more about this section of the abstract. We have described acronyms used in the abstract and throughout the document at their first use.

Reviewer: In the main text, the materials and methods section can benefit from more detail. The sampling criteria (lines 102) for choosing the regions could be explicated.

Response: Thank you. We have elaborated the text in the methods and material section (sampling criteria used in choosing regions) and made it explicit (Lines # 127 to 129). The present study is part of a larger survey that aimed to estimate the effect of an intervention called the optimizing health extension program on the utilization of child health services at the primary healthcare level in Ethiopia. The study regions and districts are selected by the Ministry of Health and Regional Health Bureaus with the intention of implementing the interventions and evaluating their effectiveness. More detail about the larger survey was found in the cited protocol paper (Lines # 132).

Reviewer: In Line 133 you should clarify if the difference you mention is the difference in proportions or scores.

Response: Thank you. The difference considered is a difference in proportions and we have added text that describes it (Lines # 32 to 33).

Reviewer: Lines 136 through 142 describe the criteria for classifying HEWs as having a know-do gap or not. I am not sure to have completely understood the relation between this and the data from direct observation of caregivers. Could you please make this clearer in the document?

Response: Thank you. Knowledge of health extension workers about the management of sick children was measured using interviews, while that of the actual practice was measured using observations of clinical consultations related to both childhood conditions (pneumonia and diarrhea). The same individual/health extension workers participated in both interviews and observation of sick child consultations. Accordingly, the know-do gap is the difference between the knowledge of health extension workers and their actual practices as measured by the observer. We have elaborated the text in the methods part to make this clearer (Lines # 147 to 151; 164 to 168).

Reviewer: Line 150: can you please clarify the “measures” that are mentioned?

Response: Thank you. The measures that were mentioned by the health extension workers are (1) prescribing treatments (ORS and Zinc), and (2) Counseling on medications (Lines # 183 to 185).

Reviewer: Line 151: Can you please clarify the reason for using the Ethiopian University Index and cut-offs?

Response: Thank you. We have used the university grading system, as there was no standard measurement to determine the level of knowledge and actual practice (skill) of health extension workers in Ethiopia. We added text that explains the reason for using the university grading index (Lines # 186 to 188).

Reviewer: Lines 156 to 159: can you clarify the criteria for “correct practice” and “correct management” and “correct treatment”.

Response: Thank you. The correct practice of providers is measured as ‘correct assessment practice’ and ‘correct treatment practice’ by comparing it with the iCCM guidelines. We have rewritten the text to make it clear in the revised manuscript (see Lines # 197 to 205).

Reviewer: Line 173: should describe the “health facility assessment data set”.

Response: Thank you. The health facility data set captures information about health posts’ infrastructure, availability of medical equipment, medicines, and job aids used in the assessment and management of sick children. We have added text and elaborated it (See Lines 233 to 235).

Reviewer: Lines 175 to 181: should detail the logistic regression model namely variables used for adjustment and in the results section you present adjusted Odds Ratios.

Response: Thank you. The variables used for adjustment/explanatory variables are the provider characteristics, iCCM training status, presence of supportive supervision and mentorships, and presence of medicines and supplies used for the management of sick children with pneumonia and diarrhea. Bivariate logistic regression was used to select candidate variables for multivariable logistic regression at $P\text{-value} < 0.2$. In the multivariable logistic regression, variables with $P\text{-value} < 0.05$ were considered statically significant. We have elaborated the text to make it clearer (See Lines # 242 to 249).

Reviewer: Lines 186 and 187: Did all participants provide a written consent? How did you manage those participants that could not read nor write?

Response: Thank you. For those caregivers who don’t read/write, a witness was presented and has appended signatures on the consent form. We have added text to make it clear (Lines # 257 to 258).

Reviewer: Lines 195 to 197: not all participants provided data on practice and knowledge. Can you specify these results?

Response: Thank you. A total of 274 health extension workers provided data on knowledge, of whom only 150 participated in the observation of clinical consultations. A description of this is provided in the manuscript (See Lines # 267 to 268).

Reviewer: Line 199: please clarify “level-3 training “, despite this is described in Table 1 but as a footnote.

Response: Thank you. The health extension workers had two levels of formal education: level-3 and level-4 education. Level-3 health extension workers are those who graduated with a certificate after a one-year pre-service training on the 17 packages of the Health Extension Program. Level-4 health extension workers are those who have level-3 certificates and got one additional year of training at a

technical college. We have described this in the revised manuscript for more clarification (see Lines # 221 to 230).

Reviewer: Line 208 - Table 1: The column heading “frequency” should be replaced by “number” or “n”, as both columns represent frequencies (absolute and relative).

Response: We have corrected it as suggested.

Reviewer: I suggest that the discussion of results could include and discuss the role of other explanatory factors such as years of practice, severity of cases, and organization of care, which may be confounding the obtained associations. I wish you all the best for your work.

Response: Thank you a lot once again for the kind wishes. As suggested we have discussed findings related to years of practice (Lines # 463 to 465; 468 to 471). However, the severity of cases is not examined in the current research and we have indicated it in the limitation of the study section of the discussion (Lines # 495 to 496). The current assessment was conducted at the community health posts within the primary healthcare level, the lowest care level in Ethiopia. At this level, the health extension workers (providers) are supposed to diagnose and treat moderate cases at their level and refer severe cases to the health centers. We have added text related to the organization of care in Lines # 486 to 488.

REVIEWER 2:

Comments to the Author:

REVIEWER 3:

Reviewer: Use APA 7th version for the references, Highlight the interventions proposed, and Mention the limitations of the study

Response: Thank you a lot. We have used the Vancouver style of referencing as it was recommended by the journal. The aim of the current analysis is to examine the know-do gap in sick child healthcare based on the end-line survey data, and as a result, we didn't provide more information about the proposed interventions. More information about the planned larger study including the interventions is briefly presented in the cited protocol paper (Lines # 132). We discussed the limitation of the study at the end of the discussion (Lines # 492 to 500).

VERSION 2 – REVIEW

REVIEWER	Dias, Carlos National Institute of Health Ricardo Jorge, Department of Epidemiology
REVIEW RETURNED	05-May-2023

GENERAL COMMENTS	Thank you for addressing the majority of the points I sent after reading the initial version of the paper. Overall, this second version now details crucial aspects of the work done. However, some aspects still need to be addressed. I will go through them and wish you may find my suggestions useful.  1. The text needs revision concerning the English language. 2. The Introduction (line 115 to line 174) section can be shortened. Several examples that support the research do not need to be described one by one. The factors involved may be summarized, as well as the expected benefits. 3. In lines 177 to 181, it should be clarified if the four study regions are the most populous regions of the country, or if other populous regions could have been selected. The text reads “...across the four populous regions...” which is not clear. In addition, you should be briefly describe the selection criteria used by the Ministry of Health, even if already published in another paper or report. 4. Please clarify the different time periods reported in lines 179 and lines 185, and its eventual implications for the methods used.
--

	5. The complex sampling strategy and sample design are described in lines 189 to 196. However this has implications in the data analysis, that area not described in the data processing and analysis section (line 301 to line 324). 6. I suggest that most of the results described in the results section could be transferred in Tables, thus reducing the length of the text and improving readability. 7. I suggest you avoid using attributive adjectives such as in line 565.
--	--

VERSION 2 – AUTHOR RESPONSE

REVIEWER 1:

Reviewer: Dear Prof Daka, thank you for addressing the majority of the points I sent after reading the initial version of the paper. Overall, this second version now details crucial aspects of the work done. However, some aspects still need to be addressed. I will go through them and wish you may find my suggestions useful. The text needs revision concerning the English language.

Response: Dear Reviewer, thank you very much for the insightful feedbacks you provided to us. We have revised the language in the revised manuscript. It is edited by language editor.

Reviewer: The Introduction (line 115 to line 174) section can be shortened. Several examples that support the research do not need to be described one by one. The factors involved may be summarized, as well as the expected benefits.

Response: Thank you. We have summarized and shortened this section of the paper. Please see the revised paper (page 5, lines # 85 and afterwards).

Reviewer: In lines 177 to 181, it should be clarified if the four study regions are the most populous regions of the country, or if other populous regions could have been selected. The text reads "...across the four populous regions..." which is not clear. In addition, you should be briefly describe the selection criteria used by the Ministry of Health, even if already published in another paper or report.

Response: Thank you. We have corrected the statement and make it clearer (Lines # 115-116). Moreover, we have also added a statement to elaborate the criteria used for the selection of study districts (Lines # 117-121).

Reviewer: Please clarify the different time periods reported in lines 179 and lines 185, and its eventual implications for the methods used.

Response: Thank you. The study period for the current paper (end-line survey) was from December 2018 to February 2019. We have described the study period for the baseline survey, just to indicate that this specific research is part of the larger survey, which incorporated both baseline and end line surveys. Mentioning the baseline survey date will not have any implications on the methods used.

Reviewer: The complex sampling strategy and sample design are described in lines 189 to 196. However, this has implications in the data analysis, that area not described in the data processing and analysis section (line 301 to line 324).

Response: Thank you. We added a statement that describes this in the data processing and analysis sections (Lines # 232 to 235).

Reviewer: I suggest that most of the results described in the results section could be transferred in Tables, thus reducing the length of the text and improving readability.

Response: Thank you. We have edited this part of the paper as suggested.

Reviewer: I suggest you avoid using attributive adjectives such as in line 565.

Response: Thank you. We have corrected as suggested.

VERSION 3 – REVIEW

REVIEWER	Dias, Carlos National Institute of Health Ricardo Jorge, Department of Epidemiology
REVIEW RETURNED	24-Jul-2023

GENERAL COMMENTS	Thank you for addressing most issues I raised after reading the second version of the paper. Overall, this third version now details crucial aspects of the work done. However, some aspects still need to be addressed. I will go through them and wish you may find my suggestions useful. 1. In the abstract, lines 28, 28, 30, 31 should be under “methodology (line 32) not before.2. The estimates in the results section of the abstract need revising. For example, the estimates of correct knowledge in lines 35 and 36 are not consistent with estimates in the results section of paper (lines 300 – 306).3. The complex sampling strategy is not addressed in this review. The resulting implications for data analysis, which may include underestimated standard errors and overestimated test statistics, are still not described in the data processing and analysis section.
---

VERSION 3 – AUTHOR RESPONSE

REVIEWER 1:

Reviewer: Dear Prof Daka, thank you for addressing most issues I raised after reading the second version of the paper. Overall, this third version now details the crucial aspects of the work done. However, some aspects still need to be addressed. I will go through them and wish you may find my suggestions useful. In the abstract, lines 28, 30, 31 should be under methodology (line 32) not before.

Response: Dear Reviewer, thank you very much for the helpful feedbacks you provided to us. We have reformatted the abstract and revised this part of the paper.

Reviewer: The estimates in the results section of the abstract need revising. For example, the estimates of correct knowledge in lines 35 and 36 are not consistent with estimates in the results section of the paper (lines 300-306).

Response: Thank you. The results presented in the abstract was based on the analysis performed among 150 HEWs, the sample size used for know-do gap analysis. The abstract section of the result was taken from Line # 307 and afterwards. Lines 300-305, has presented the knowledge findings based on an analysis of 274 HEWs.

Reviewer: The complex sampling strategy is not addressed in this review. The resulting implications for data analysis, which may include underestimated standard errors and overestimated test statistics are still not described in the data processing and analysis section.

Response: Thank you. We have added statement and described the resulted implications in the data processing and analysis section (Lines # 230 to 234). Moreover, we have included information in the limitations section of the paper under discussion (Line # 467).